# Demonstration of targeted crossovers in hybrid maize using CRISPR technology

Andrei Kouranov [1][✉], Charles Armstrong[1], Ashok Shrawat[1], Vladimir Sidorov[1], Scott Huesgen[1], Bryce Lemke[1], Timothy Boyle[1], Michelle Gasper[1], Richard Lawrence[1] & Samuel Yang[1]

Naturally occurring chromosomal crossovers (CO) during meiosis are a key driver of genetic diversity. The ability to target CO at specific allelic loci in hybrid plants would provide an advantage to the plant breeding process by facilitating trait introgression, and potentially increasing the rate of genetic gain. We present the first demonstration of targeted CO in hybrid maize utilizing the CRISPR Cas12a system. Our experiments showed that stable and heritable targeted CO can be produced in F1 somatic cells using Cas12a at a significantly higher rate than the natural CO in the same interval. Molecular characterization of the recombinant plants demonstrated that the targeted CO were driven by the non-homologous end joining (NHEJ) or HDR repair pathways, presumably during the mitotic cell cycle. These results are a step towards the use of RNA-guided nuclease technology to simplify the creation of targeted genome combinations in progeny and accelerate breeding.

---

[1] Bayer Crop Science, Chesterfield, MO, USA. [✉]email: andrei.kouranov@bayer.com

Improvement of genetic gain to increase yield while maintaining genetic diversity are key fundamentals of plant breeding[1]. Variation in genetic gain and diversity are introduced by homologous recombination (HR) resolved by CO during meiosis. Naturally occurring CO are rare events, introduced at a frequency of about one to three CO per chromosome pair in meiotic cells[2,3]. Targeted CO at specific genomic loci could facilitate crop improvement by breaking linkage drag and/or stacking haplotypes associated with high yield. Rex Bernardo predicted that targeted CO at prescribed loci in hybrid maize could considerably improve genetic gain and increase yield. The author's analysis showed that maize grain yield could be doubled if one targeted CO was introduced into each chromosome[4]. A similar effect of targeted CO on average yield gain was predicted in other crop species including soybean, wheat, barley, and pea[5]. Plant breeders rely on natural CO, creating and screening large populations of biparental crosses to identify and stabilize the desired genotypes for crop improvement. However, this approach is time-consuming, expensive, and limited due to low CO frequencies at some genomic locations. CRISPR-Cas technology provides an alternative genome engineering approach to implement targeted CO and accelerate the process of breeding. Several advancements in precision genome engineering have already been demonstrated using this technology in plants, including homologous chromosome recombination[6], reciprocal trans-fragment translocations[7], and large fragment inversions[8,9]. In addition, the guided nuclease technology can be employed to induce homologous recombination in somatic cells, avoiding competition with the naturally occurring process during meiosis[6].

In this study, we demonstrated targeted, heritable COs between parental chromosomes in hybrid maize. To identify COs in maize hybrid, we employed automated, high-throughput seed chipping technology followed by genotyping with multiple TaqMan assays. We supported our observation by two independent genome editing experiments. Our experiments confirm that targeted DSBs introduced in somatic cells can lead to CO that is stable, transmissible to gamete cells, and heritable in the next generation.

## Results

### Experimental design and workflow to demonstrate targeted crossover.
To evaluate whether targeted CO can be directed by guided nuclease activity, we developed an experimental strategy to induce allelic CO by introducing double-stranded breaks (DSB) at the same position in both parental chromosomes in hybrid somatic cells. As a guided nuclease, we employed LbCas12a whose expression was driven by the strong constitutive Zm-Ubiquitin1 promoter[10]. Two gRNA target sites about 180 Kb apart, located between 170.9 and 173.1 cM on chromosome 3 were selected for targeted recombination (Fig. 1).

The targeted CO at the selected target sites were assayed using 18 and 20 TaqMan markers, respectively (M1-M17, and M33 for the gRNA1 target site; M3, M16-M34 for the gRNA2 target site; Fig. 1). The assayed SNPs (Single-Nucleotide Polymorphism) were distributed across 556 kilobases (Kb) (M1-M33) and 359 Kb (M3-M34) regions. The most distant markers were located ~290 Kb, 140 Kb, 70 Kb, and 6 Kb on each side from the gRNA target site (Supplementary Data 3). In order to increase confidence that the targeted recombination was introduced precisely at the gRNA target sites, we designed TaqMan assays located within a 6 Kb region spanning the recombination target sites (M4-M15 spanning gRNA1 and M19-M31 spanning gRNA2 target sites). The closest TaqMan markers were located about 400 nt and 200 nt on each side of the targeted DNA break introduced by gRNA1 and gRNA2, respectively (Fig. 1: Marker 8 and 9, and Marker 25 and 26). Both gRNAs showed strong editing activity in the F1-T0 plants. An average number of reads with the DSB editing patterns was 87 and 80% at the gRNA1 and gRNA2 target sites, respectively (Supplementary Fig. 1a, b). All assayed F1-T0 plants contained multiple patterns of DSB editing showing chimerism frequently observed in primary (T0) plant transformants[11,12]. Distribution of the editing patterns in the F1-T0 plants is shown in four examples (Supplementary Fig. 2a, Supplementary Fig. 2b, Supplementary Data 1). We cannot definitively conclude that the editing observed in the F1-T0 plants is mono or biallelic because of the absence of the allelic polymorphism at the gRNA1 and gRNA2 target sites. However, we can assume that both parental alleles were edited in plants where editing efficiency was above 50%.

Two independent plant transformation experiments were performed using T-DNA plasmids containing the guided nuclease and either gRNA1 or gRNA2 to evaluate the efficiency of targeted CO at two distinct chromosomal positions. First, F1 hybrid plants were produced by crossing two elite maize inbred lines (Parent A or Parent B) (Fig. 2a). Next, embryo explants were isolated from mature F1 hybrid seeds and separated into editing and control treatments for Agrobacterium-mediated transformation (Fig. 2b). The targeting gRNAs produced DSB at the desired recombination locations, whereas the control gRNA was lacking the target sites, and thus no editing was observed. The regenerated F1-T0 plants were reciprocally backcrossed to one of the parents for the identification of chromosomal recombination (Fig. 2c, d). A small amount of endosperm tissue was non-destructively isolated from the harvested BC1-F1 seeds for genotyping analysis (Fig. 2e) to

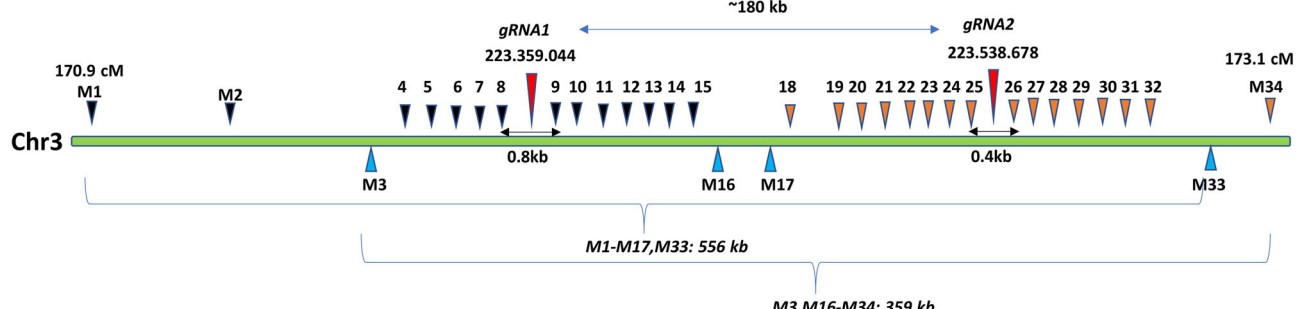

**Fig. 1 Schematic diagram of the two target regions (gRNA1 and gRNA2) and the locations of the polymorphic SNP markers (M1-M34) used in the genotyping assays.** Schematic SNP marker positions are shown as black, brown, and blue triangles. The blue triangles represent SNP markers that were shared in the two different genome editing experiments. The physical genome coordinates of the gRNA target sites are based on the B73 genome reference public assembly: Zm-B73-REFERENCE-NAM-5.0. (https://www.maizegdb.org)[11]. The physical position of each SNP marker can be found in Supplementary Data 3.

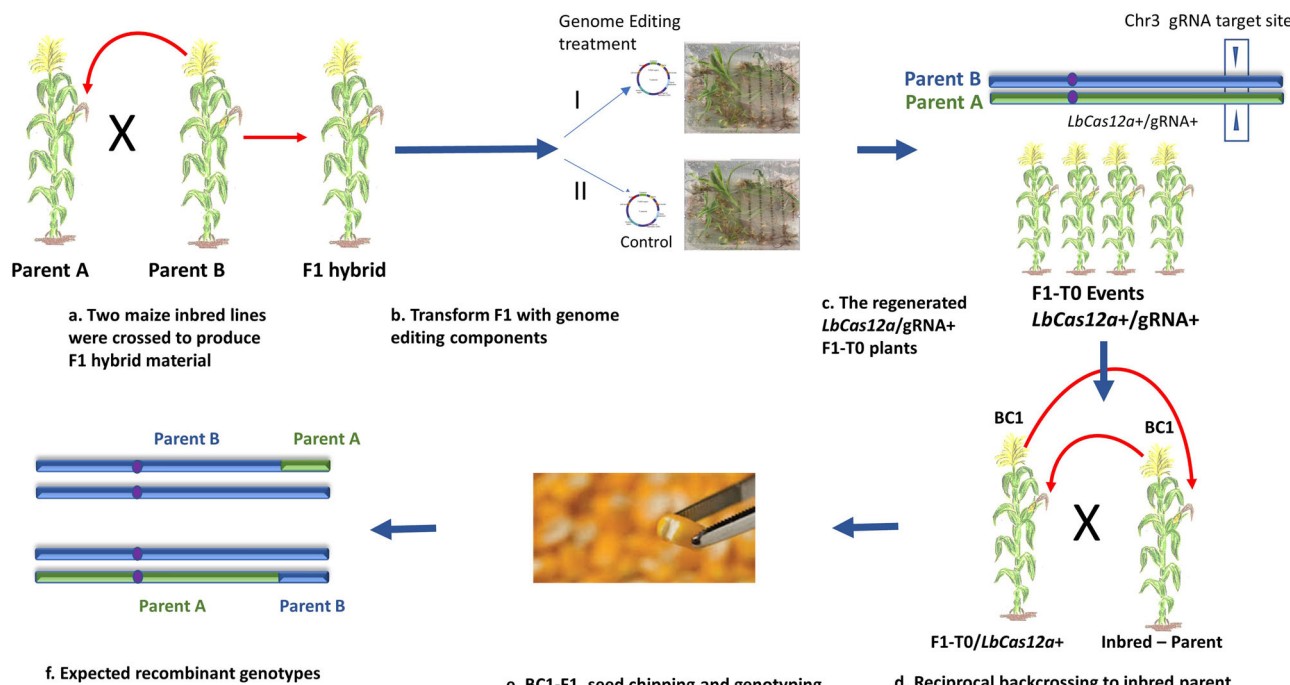

**Fig. 2 Experimental workflow to demonstrate guided homologous chromosome recombination in maize. a** Two maize inbred lines were crossed to produce F1 hybrid material. **b** Isolated F1 embryo explants were separated into two groups for *Agrobacterium* -mediated transformation: (I) editing treatment; (II) control treatment. **c**, **d** The regenerated F1-T0 plants were reciprocally backcrossed to Parent B (the first experiment: gRNA1) or Parent A (the second experiment: gRNA2) for the identification of CO by genotyping. **e** The harvested BC1-F1 seeds were chipped to isolate a small amount of endosperm tissue for genotyping analysis using SNP TaqMan assays. **f** The genotyping results were examined for the presence of the expected reciprocal chromosomal recombination between the parental chromosomes.

**Table 1 Result of the genotyping analysis and identification of the BC1-F1 seeds with chromosomal recombination at the gRNA1 target site.**

| Constructs | Transformation | Events# | Total seeds genotyped | Seeds with recombinations between M1-M33 556 kb region (%) | Seeds with recombinations between M8-M9 0.8 kb region (%) |
|---|---|---|---|---|---|
| pCpf1_gRNA1 | Editing plasmid | 42 | 4200 | 135 (3%) | 30 (0.71%) |
| pCpf1_gRNA_control | Control plasmid | 15 | 2265 | 69 (3%) | 1 (0.04%) |

identify seeds with reciprocal CO between the parental chromosomes (Fig. 2f).

**Identification of the targeted CO at the gRNA1 target site**. In the first guided CO experiment 42 edited and 15 control F1-T0 plants were selected for reciprocal backcrossing with inbred Parent B. Subsets of BC1-F1 seeds from each backcross event were sampled for genotyping analysis (Table 1). Seeds from both reciprocal backcrosses were included in the analysis (Supplementary Table 1). The DNA extracted from the seed tissue was genotyped using 18 TaqMan PCR assays (Fig. 1, M1-M17, M33). We observed a significant difference in the recombination frequency between the edited and control BC1-F1 seed populations in the 0.8 Kb (between M8 and M9) region spanning the guided nuclease target site (Table 1). Within this narrow interval, overall recombination frequency in the edited population was 0.71% (30 out of 4200 genotyped seeds) versus 0.04% (1 out of 2265 genotyped seeds) in the control, a significant increase of ~18-fold in the presence of the guided nuclease. The Chi-square test of independence validated by the permutation test confirmed that the observed difference in recombination frequency between the control and treatment is statistically significant. The probability that the frequency of recombination in the edited population is independent of guided nuclease activity is <0.1% ($p$-value <

0.001). However, we did not observe an overall increase in recombination frequency between the edited and control populations within the larger 556 Kb (between M1 & M33) interval (Table 1). The observed 3% recombination frequency within this interval is consistent with previously defined natural recombination rates, suggesting a precise and specific increase in targeted recombination at the gRNA1 target site.

Recombinant chromosomes from both reciprocal CO were identified by genotyping (Supplementary Fig. 3a). About 43% of the recombinant BC1-F1 seeds (13 of 30) from edited plants with CO were LbCas12a negative (Table 2, and Supplementary Fig. 3a), indicating germinal transmission of the recombinant chromosome from the F1-T0 plant to the next generation.

Our analysis showed that BC1-F1 seeds with the recombinant genotype derived from only three edited F1-T0 plants (Table 2: Event 1, Event 2, Event 3). We genotyped all seeds derived from these three edited and three control F1-T0 plants and compared the recombination frequencies (Table 2). One of the control plants was selected for the comparison because it contained one seed with CO within the 0.8 Kb targeted region (Table 2 and Supplementary Fig. 3a: Event 4). Two other control seed sets were selected randomly (Event-5 and Event-6). The rate of targeted CO in two of the three edited plants ranged from 3.5–6.0% versus 0.1% in the control (Table 2). These editing rates are consistent

**Table 2 The BC1-F1 seeds with chromosomal recombination identified at the gRNA1 target site grouped by F1-T0 plants (events).**

| Transformation | Event # | Total seeds genotyped | Seeds with recombinations between M1-M33 556 kb region (%) | Seeds with recombinations between M8-M9 0.8 kb region (%) | LbCas12a negative seeds |
|---|---|---|---|---|---|
| pCpf1_gRNA1 | Event 1 | 308 | 27 (9%) | 18 (6%) | 8 |
| pCpf1_gRNA1 | Event 2 | 311 | 20 (6.5%) | 11 (3.5%) | 4 |
| pCpf1_gRNA1 | Event 3 | 461 | 19 (4%) | 1 (0.2%) | 1 |
| pCpf1_gRNA_control | Event 4 | 542 | 13 (2.4%) | 1 (0.1%) | |
| pCpf1_gRNA_control | Event 5 | 543 | 15 (3%) | 0 | |
| pCpf1_gRNA_control | Event 6 | 196 | 1 (0.5%) | 0 | |

**Table 3 Result of the genotyping analysis and identification of the BC1-F1 seeds with chromosomal recombination at the gRNA2 target site.**

| Constructs | Transformation | Event# | Total seeds genotyped | Seeds with recombination between M3-M34 359 Kb region | Seeds with recombination between M25-M25 0.4 Kb region |
|---|---|---|---|---|---|
| pCpf1_gRNA2 | Editing plasmid | 43 | 4798 | 197(4%) | 175(3.6%) |
| pCpf1_gRNA_control | Control plasmid | 15 | 3370 | 37(1%) | 0 |

with previously reported genome editing rates in plants[13,14]. The single recombinant seed identified in Event-3 may potentially represent a natural CO. All recombinant BC1-F1 seeds listed in Table 2 derived from backcrosses where F1-T0 plants were used as females suggesting that guided CO transmitted to the female gamete before pollination with the inbred pollen. We were not able to identify examples of the guided CO in the male gametes in this experiment for Event-1, Event-2, or Event-3. However, this may be due to the lower number of genotyped seeds from backcrosses as male versus female (173 versus 887 seeds, see Supplementary Table 2 for additional details).

**Identification of the targeted CO at the gRNA2 target site**. In our second genome editing experiment, we intended to confirm that CO in somatic cells could be targeted at a different gRNA site and transmitted through the male gametes. We produced and backcrossed 43 F1-T0 plants (Table 3) as males onto Parent A. The total number of edited and control BC1-F1 seeds genotyped in this experiment are shown in Table 3. We used a subset of seeds from the same negative control population used in the analysis of the first genome editing experiment to evaluate and compare the efficiency of targeted and natural CO. Similar to the edited BC1-F1 seed population, all control seeds were produced by crossing the control F1-T0 plants as males onto inbred females. We employed 20 TaqMan PCR assays to evaluate targeted recombination at the gRNA2 target site (M3, M16-M34, Fig. 1). In this experiment, we observed a higher recombination rate in the edited versus control BC1-F1 seed populations within the 359 Kb and 0.4 Kb regions (Table 3). The recombination rate within the 359 Kb region (between M3 and M34) was 4-fold higher in the edited than the control seed population. Remarkably, most of the observed targeted CO were within the narrow 400 nt region between M25 and M26, representing a rate of 3.6%. We did not identify any seeds in the control population with chromosomal recombination between the M25 and M26 markers. All seeds with a targeted CO (175 seeds total) were derived from a single F1-T0 plant: ZM_S22440456 (Table 4). This result indicates that a guided CO was introduced into an early meristematic precursor cell and transmitted to the developing gametes. Both reciprocal chromosomal COs were identified in the ZM_S22440456 BC1-F1 seed population in about equal proportions (Table 4; 79 and 96 seeds) Examples of both reciprocal CO are shown in Supplementary Fig. 3b.

**Analysis of the DNA editing patterns at the sites of the targeted CO**. In order to further characterize the precise nature of the targeted CO at the gRNA target sites, we selected all LbCas12a-negative BC1-F1 seeds with targeted CO from the first genome editing experiment with gRNA1 (13 total seeds across: Event-1, 2, and 3. Fig. 3b, Supplementary Data 1) and 14 randomly selected LbCas12a-negative BC1-F1 seeds with the targeted CO from the single event identified with the targeted CO in the second experiment with gRNA2 (Fig. 3d, Supplementary Data 1) for further analysis. Re-genotyping of leaf tissue from young plants from these 27 seeds confirmed the targeted CO and the absence of the LbCas12a cassette. To analyze the editing patterns at the site of the targeted CO, gRNA target sites were amplified and sequenced using Illumina technology. As non-edited controls, we included samples from Event-4 plants from the first genome editing experiment, one of which showed natural CO between M8 and M9 SNP markers. Our analysis of the BC1-F1 plants from the first genome editing experiment showed that in 5 out of the 13 analyzed edited plants, the DNA breaks were repaired without any mutations (Events 1.4, 1.8, 1.9, 2.10, 2.11, Table 5). Error-prone NHEJ is the most prevalent DNA-repair pathway in plant somatic cells. This mechanism may introduce deletions or insertions at the site of DSB repair, but it may also be repaired without any mutations[15,16]. We also cannot exclude that the observed accurate DNA repair was driven by the HDR pathway. Only one DNA editing pattern was identified in each BC1-F1 plant sample, which is consistent with the stability of the targeted CO transmission in nuclease-negative progeny (Table 5 and Table 6). Interestingly, DNA editing patterns at the targeted CO site differed across BC1-F1 progeny derived from Event-1 and Event 2. In eight examined Event-1 siblings, we observed four different DNA repair patterns (Table 5). This result is consistent with mosaic editing outcomes frequently observed in the presence of CRISPR-mediated nucleases in plants[11]. The mosaicism can be explained by the editing machinery expression delay relative to the meristematic cell division. A typical corn ear contains 400–500 kernels that develop from different meristematic precursor cells[17]. The germline fate of the somatic precursors is defined late in plant development after several cell divisions, leading to variation in DNA repair patterns[18]. Targeted CO with the repair without any change at the target sequence could also be subjected to recurrent editing after additional cell divisions.

**Table 4 The BC1-F1 seeds with chromosomal recombination at the gRNA2 target site derived from a single F1-T0 event.**

| Transformation | Event | Seeds with B/A recombination | Seeds with A/B recombination | Total seeds genotyped | Seeds with B/A recombination LbCas12a negative | Seeds with A/B recombination LbCas12a negative | Total Seeds LbCas12a negative |
|---|---|---|---|---|---|---|---|
| pCpf1_gRNA2 | ZM_S22440456 | 79 | 96 | 175 | 34 | 57 | 91 |

In contrast to the BC1-F1 plants edited with gRNA1, all analyzed plants from the second genome editing experiment with gRNA2 showed a single editing pattern (Table 6). This result confirms that targeted CO was introduced in the early somatic progenitor cell that was transmitted to developing gametes.

**Isolation of homozygous recombinant plants and confirmation of the heritability of the targeted CO in hybrid maize.** To confirm the stability of the targeted CO, we selected three LbCas12a-negative BC1-F1 seeds heterozygous for the chromosome recombination (Event-1.18, Event-2.10, and Event-2.11: Fig. 3b, Supplementary Data 1). The plants were self-pollinated to produce a BC1-F2 segregating population (Fig. 4a). The segregating populations of the one-week-old BC1-F2 seedlings were sampled and genotyped with 18 TaqMan markers. The genotyping analysis identified individual plants with the targeted CO (Fig. 4b: labeled Light Green and Blue, Supplementary Data 1) at the expected segregation ratio of 1 homozygous CO:2 heterozygous CO:1 homozygous wild-type (no CO) in all three tested segregating populations. Chromosome recombination in this study was stable and did not cause any negative impact on plant growth or development. The analysis of the guided CO sites in the homozygous recombinant BC1-F2 plants with Illumina sequencing confirmed previously observed DNA editing patterns (compare Table 5 and Fig. 4c, Supplementary Data 1). BC1-F2 siblings from Event-1.18 showed a deletion at the point of the guided CO presumably driven by short nucleotide microhomology (Fig. 4c, highlighted in gray), whereas in siblings from Event-2.10 & 11 DNA break was repaired without any mutation.

We performed a similar analysis to confirm stable transmission of the targeted CO introduced at the gRNA2 target site. Three BC1-F2 seed populations (84–88 seeds total) derived from BC1-F1 events (Table 6; ZM_S22440456-1 to ZM_S22440456-3) were genotyped with 20 TaqMan markers. The analysis confirmed the presence of the targeted CO in the expected 1:2:1 segregation ratio in all three tested BC1-F2 populations. The genotyping results for the BC1-F2 seed population derived from event ZM_S22440456-1 are provided in Supplementary Data 4 as an example.

## Discussion

We demonstrated guided CO in F1 hybrid maize in two independent experiments using a guided LbCas12a nuclease. Both experiments confirmed that a transgenically introduced guided nuclease can effectively drive targeted CO in somatic maize cells transmittable to the next generation. The results of the two genome editing experiments were different. Both tested editing reagents (gRNA1 & gRNA2) showed high DNA editing activity ranging from 30–100% as evaluated by Illumina sequencing (Supplementary Fig. 1a, Supplementary Fig. 1b, Supplementary Data 1). A high frequency of targeted CO should be expected in the backcross progeny if active gRNAs induce a targeted CO in the germline precursor cells early in plant development. This result was demonstrated in our second genome editing experiment with gRNA2. We identified one out of forty-three tested F1-T0 plants in which the targeted CO was transmitted to all backcrossed progeny (Table 4). Analysis of the DNA editing at the site of the CO in all tested LbCas12a negative BC1-F1 plants identified a single editing pattern (Table 6). These results are consistent with the hypothesis that a targeted CO was introduced in an early germinal progenitor cell that was transmitted to all gametes in the F1-T0 plant.

The genome editing experiment with gRNA1 produced three F1-T0 plants with targeted CO. However, targeted CO was detected only in a subset of seeds showing the CO transmission

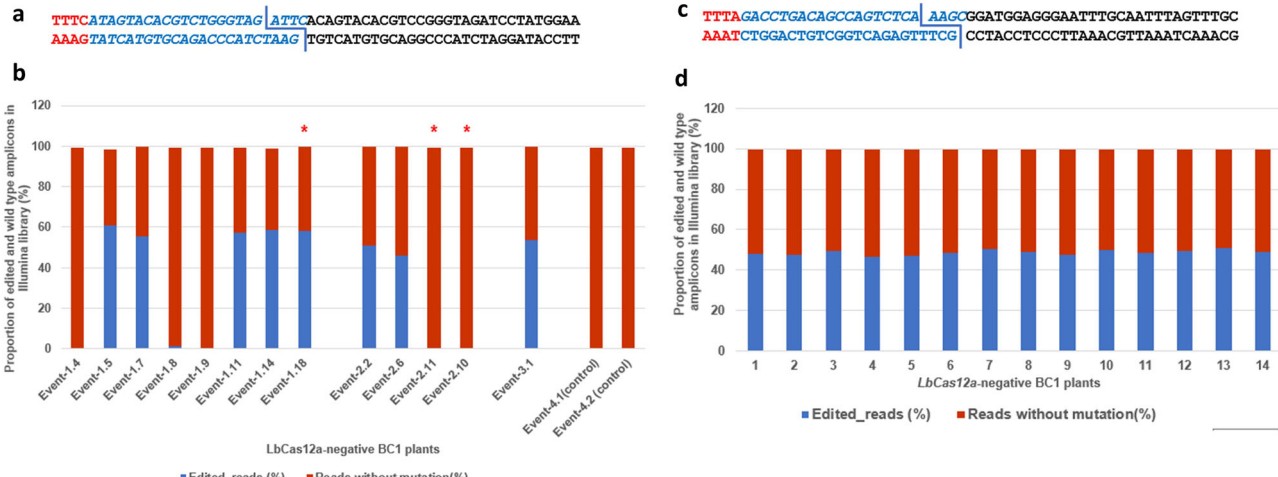

**Fig. 3 Analysis of DNA editing patterns at the sites of targeted CO in the BC1-F1 plants identified in the genome editing experiments with gRNA1 and gRNA2 using Illumina sequencing.** DNA was isolated from a single leaf disc sampled from each BC1-F1 plant for library construction and sequencing. Each bar on the graph represents a result derived from a single BC-F1 plant (*n* = 1). **a** The gRNA1 target site showing *LbCas12a* cutting pattern: the PAM sequence (red); gRNA target sequence (italic blue); a fragment of the downstream DNA sequence (black). **b** The graph shows the proportion of edited (blue) and non-edited (red) reads in 13 *LbCas12a* negative BC1-F1 plants with the targeted CO and two control plants. BC1-F1 plants selected for further characterization are highlighted with red stars (*) **c** The gRNA2 target site showing *LbCas12a* cutting pattern; **d** The graph shows the proportion of edited (blue) and non-edited (red) reads in 14 randomly selected *LbCas12a* negative BC1-F1 plants with targeted CO derived from F1-T0 by Parent A backcross.

rates in each F1-T0 plant as 0.2, 3.5, and 6.5%, respectively (Table 2). This is significantly lower than in the example with gRNA2 where the targeted CO transmission was 100%. We also observed mosaicism in the DNA editing patterns at the targeted CO sites across examined LbCas12a negative BC1-F1siblings using Illumina sequencing. In eight examined Event-1 and four Event-2 siblings, we observed four and two different DNA repair patterns, respectively (Table 5). These results are consistent with the hypothesis that in the genome editing experiment with gRNA1 the targeted CO were introduced in the germinal tissue at a later stage in plant development. Analysis of the DNA repair patterns at the gRNA1 target site in F1-T0 plants sampled at the V1-growth stage showed the presence of unedited target sites ranging from 0.3 to 54% indicating that some somatic cells maintained one or both wild type alleles until later in development (Supplementary Fig. 1a, Supplementary Fig. 1b, Supplementary Data 1). Preservation of unedited alleles until the later stages of plant development can be explained by the activity of the c-NHEJ (Canonical-NHEJ) DNA repair pathway. The c-NHEJ repair pathway can result in perfect repair of the DSBs making the gRNA target sites available for editing in the next cell generations[13,14]. Constitutive expression of the LbCas12a nuclease can induce editing of the intact target sites at any stage of plant growth and development[19,20] which can contribute to the observed mosaicism. We assume that the targeted CO in the experiment with gRNA1 could be induced after the subepidermal cells of the ovule primordia developed into the archesporial cells that represent precursors for the MMC (Megaspore Mother Cell), or maybe even later when the MMC undergoes meiosis[21]. Independent genome editing events in the somatic cell lineages developing to 400–500 maize ovaries can explain mosaic editing in our experiment with gRNA1.

Targeted CO were detected in 7% (three events out 42 tested) and 2.3% (one event out 43 tested) of the F1-T0 plants in our genome editing experiments using gRNA1 and gRNA2, respectively. Illumina sequencing of the target site amplicons in F1-T0 plants showed that the majority of the targeted DSBs were repaired by direct re-ligation of the broken DNA ends. The efficiency of targeted CO can be affected by the asynchronous

editing of the homologous target sites. If one of the homologous target sites is mutated it becomes unavailable for chromosomal rearrangement driven by the NHEJ repair pathway. In certain examples, the asynchronous DSB in one of the chromosomes could be repaired by the HDR pathway where the intact homologous chromosome is used as a repair template. We identified five BC1-F1 plants with targeted CO without any change in the DNA sequence at the target site, indicating that chromosomal recombination could be driven by the HDR mechanism. The frequency of the targeted CO did not show a strong correlation with gRNA editing activity in our experiments. In the first editing experiment, the highest number of BC1-F1 seeds with the targeted CO (18 seeds) was observed in the F1-T0 Event-1 where gRNA editing activity was the lowest (45%) across the analyzed F1-T0 population (Supplementary Fig. 1a, Supplementary Data 1). In the second genome editing experiment, the targeted CO was identified in a F1-T0 plant (ZM_S22440456) where the gRNA2 editing activity was below the population average (60% versus 83%) (Supplementary Fig. 1b, Supplementary Data 1). Our experiments don't provide enough statistical power to define a specific threshold of the gRNA editing activity that would facilitate the optimum frequency of the targeted CO. There were very few plants in the analyzed F1-T0 populations where gRNA editing activity was below 50%, one plant in experiment 1, and two plants in experiment 2, respectively (Supplementary Fig. 1a, Supplementary Fig. 1b, Supplementary Data 1).

Our study represents an initial step in genome engineering utilizing CRISPR-associated nuclease technology with the potential to improve genetic gain in commercial crops. We present two possible results that experimenters may observe when using genome editing reagents to induce targeted CO in the maize genome. Complex chromosomal rearrangements, such as a targeted CO, is a less frequent repair outcome comparing to direct re-ligation of the broken DNA ends driven by the NHEJ pathway. The editing reagents stably integrated into the plant genome remain active throughout different stages of plant growth and development. Induction of the targeted CO in a germline precursor cell early in development will increase the transmission rate to the next generation. The differences in the targeted CO

**Table 5 Specific DNA editing patterns introduced by the gRNA1 at the site of the targeted CO in the analyzed BC1-F1 plants.**

| BC1-F1 plant # | Construct | Edited reads (%) | Reads without mutation (%) | Editing pattern |
|---|---|---|---|---|
| Event-1.4 | pCpf1_gRNA1 | 0 | 99 | *TTTC (ATAGTACACGTCTGGGTAGATTC)* / ACAGTACACGTCCGGGTAGATCCTATGGAA |
| Event-1.5 | pCpf1_gRNA1 | 61 | 38 | *TTTC (A------------------------)* / )-CAGTACACGTCCGGGTAGATCCTATGGAA |
| Event-1.7 | pCpf1_gRNA1 | 55 | 44 | *TTTC (ATAGTACACGTCTG---------)* / ACAGTACACGTCCGGGTAGATCCTATGGAA |
| Event-1.8 | pCpf1_gRNA1 | 1 | 98 | *TTTC (ATAGTACACGTCTGGGTAGATTC)* / ACAGTACACGTCCGGGTAGATCCTATGGAA |
| Event-1.9 | pCpf1_gRNA1 | 0 | 99 | *TTTC (ATAGTACACGTCTGGGTAGATTC)* / ACAGTACACGTCCGGGTAGATCCTATGGAA |
| Event-1.11 | pCpf1_gRNA1 | 57 | 42 | *TTTC (ATAGTACACGTC-----------)* / )----------CGGGTAGATCCTATGGAA |
| Event-1.14 | pCpf1_gRNA1 | 59 | 40 | *TTTC (A------------------------)* / )-CAGTACACGTCCGGGTAGATCCTATGGAA |
| Event-1.18 | pCpf1_gRNA1 | 58 | 41 | *TTTC (A------------------------)* / )-CAGTACACGTCCGGGTAGATCCTATGGAA |
| Event-2.2 | pCpf1_gRNA1 | 51 | 49 | *TTTC (ATAGTACACGTCTCGGGTAG---C)* / ACAGTACACGTCCGGGTAGATCCTATGGAA |
| Event-2.6 | pCpf1_gRNA1 | 46 | 54 | *TTTC (ATAGTACACGTCTCGGGTAG---C)* / ACAGTACACGTCCGGGTAGATCCTATGGAA |
| Event-2.10 | pCpf1_gRNA1 | 0 | 99 | *TTTC (ATAGTACACGTCTGGGTAGATTC)* / ACAGTACACGTCCGGGTAGATCCTATGGAA |
| Event-2.11 | pCpf1_gRNA1 | 0 | 99 | *TTTC (ATAGTACACGTCTGGGTAGATTC)* / ACAGTACACGTCCGGGTAGATCCTATGGAA |
| Event-3.1 | pCpf1_gRNA1 | 54 | 46 | *TTTC (ATAGTACACGTCTGG---------)* / )------CGTCCGGGTAGATCCTATGGAA |
| Event-4.1 | pCpf1_gRNA_control | 0 | 99 | *TTTC (ATAGTACACGTCTGGGTAGATTC)* / ACAGTACACGTCCGGGTAGATCCTATGGAA |
| Event-4.2 | pCpf1_gRNA_control | 0 | 99 | *TTTC (ATAGTACACGTCTGGGTAGATTC)* / ACAGTACACGTCCGGGTAGATCCTATGGAA |

In the editing pattern column, the sequence starts with PAM motif (TTTC), the intact or mutated gRNA targeting sequences are shown in italic.

**Table 6 Specific DNA editing patterns introduced by the gRNA2 at the site of the targeted CO in the analyzed BC1-F1 plant.**

| BC1-F1 Plant # | Construct | Edited reads (%) | Reads without mutation (%) | Editing pattern |
|---|---|---|---|---|
| ZM_S22440456-1 | pCpf1_gRNA2 | 48 | 52 | *TTTA (GACCTGACAGCCAGTC–AAAGC)*<br>*GGATGGAGGGAATTTGCAATTTAGTTTGC* |
| ZM_S22440456-2 | pCpf1_gRNA2 | 48 | 52 | *TTTA (GACCTGACAGCCAGTC–AAAGC)*<br>*GGATGGAGGGAATTTGCAATTTAGTTTGC* |
| ZM_S22440456-3 | pCpf1_gRNA2 | 49 | 51 | *TTTA (GACCTGACAGCCAGTC–AAAGC)*<br>*GGATGGAGGGAATTTGCAATTTAGTTTGC* |
| ZM_S22440456-4 | pCpf1_gRNA2 | 47 | 53 | *TTTA (GACCTGACAGCCAGTC–AAAGC)*<br>*GGATGGAGGGAATTTGCAATTTAGTTTGC* |
| ZM_S22440456-5 | pCpf1_gRNA2 | 47 | 53 | *TTTA (GACCTGACAGCCAGTC–AAAGC)*<br>*GGATGGAGGGAATTTGCAATTTAGTTTGC* |
| ZM_S22440456-6 | pCpf1_gRNA2 | 48 | 51 | *TTTA (GACCTGACAGCCAGTC–AAAGC)*<br>*GGATGGAGGGAATTTGCAATTTAGTTTGC* |
| ZM_S22440456-7 | pCpf1_gRNA2 | 50 | 50 | *TTTA (GACCTGACAGCCAGTC–AAAGC)*<br>*GGATGGAGGGAATTTGCAATTTAGTTTGC* |
| ZM_S22440456-8 | pCpf1_gRNA2 | 49 | 51 | *TTTA (GACCTGACAGCCAGTC–AAAGC)*<br>*GGATGGAGGGAATTTGCAATTTAGTTTGC* |
| ZM_S22440456-9 | pCpf1_gRNA2 | 47 | 52 | *TTTA (GACCTGACAGCCAGTC–AAAGC)*<br>*GGATGGAGGGAATTTGCAATTTAGTTTGC* |
| ZM_S22440456-10 | pCpf1_gRNA2 | 50 | 50 | *TTTA (GACCTGACAGCCAGTC–AAAGC)*<br>*GGATGGAGGGAATTTGCAATTTAGTTTGC* |
| ZM_S22440456-11 | pCpf1_gRNA2 | 49 | 51 | *TTTA (GACCTGACAGCCAGTC–AAAGC)*<br>*GGATGGAGGGAATTTGCAATTTAGTTTGC* |
| ZM_S22440456-12 | pCpf1_gRNA2 | 49 | 50 | *TTTA (GACCTGACAGCCAGTC–AAAGC)*<br>*GGATGGAGGGAATTTGCAATTTAGTTTGC* |
| ZM_S22440456-13 | pCpf1_gRNA2 | 51 | 49 | *TTTA (GACCTGACAGCCAGTC–AAAGC)*<br>*GGATGGAGGGAATTTGCAATTTAGTTTGC* |
| ZM_S22440456-14 | pCpf1_gRNA2 | 49 | 51 | *TTTA (GACCTGACAGCCAGTC–AAAGC)*<br>*GGATGGAGGGAATTTGCAATTTAGTTTGC* |

In the editing pattern column, the sequence starts with PAM motif (TTTA), the mutated gRNA targeting sequences are shown in italic.

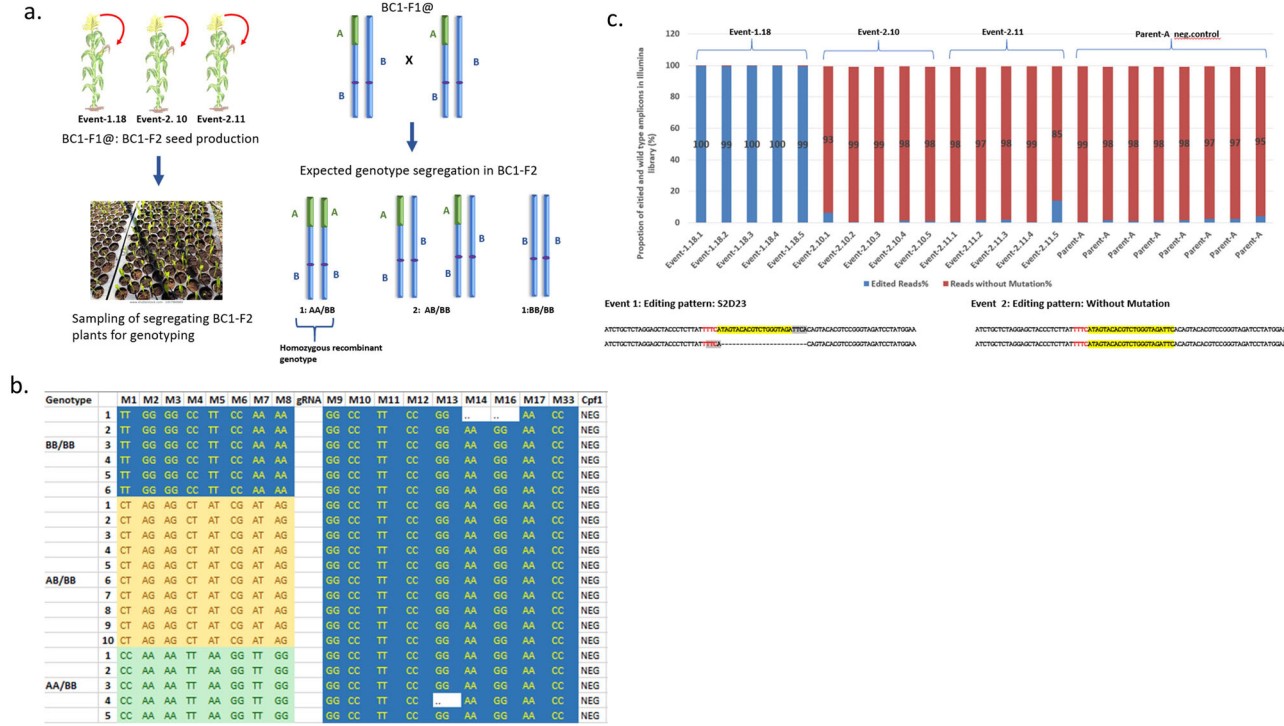

**Fig. 4 Segregation analysis of BC1-F2 populations and identification of homozygous recombinant plants. a** Schematic diagram of the segregation analysis of the BC1-F2 populations; **b** The result of the genotyping analysis of the BC1-F2 segregating population. An example of the segregation analysis of the SNP markers in 21 BC1-F2 plants from Event-2.11 is shown; **c** Confirmation of DNA editing pattern at the site of the targeted CO in homozygous BC1-F2 plants with Illumina sequencing. DNA was isolated from a single leaf disc sampled from each BC1-F2 plant for library construction and sequencing. Each bar on the graph represents a result derived from a single BC-F2 plant ($n = 1$). The proportion of edited (blue) and non-edited (red) amplicons in homozygous recombinant plants from three evaluated populations is shown on the graph. The pictures under the graph represent alignments of the sequences at the site of targeted CO with the inbred reference. gRNA1 target sequence is highlighted in yellow. The gray highlights nucleotides microhomology that could contribute to the repair of DSBs. S2d23 is an abbreviated description of the DNA editing pattern. S2 indicates the nucleotide position where the deletion starts; d23 indicates the length of the deleted nucleotides.

transmission rate observed in our two experiments could be explained by stochasticity. We can speculate that if we increased tested population size, we could observe examples of early induction of the targeted CO driven by gRNA1 activity. Another explanation could be that the gRNA1 target site is not accessible to the guided nuclease at the early stages of plant regeneration. It is known that the activity of the guided nuclease is inhibited in the compact chromatin regions[22–24]. The chromatin modifications including histone and DNA methylation that modulate chromatin compactness are developmentally regulated[25]. Changes in the target site accessibility for gRNAs during different stages of plant development can influence genome editing efficiency.

Multiplexed approaches inducing targeted CO at several homologous chromosomes simultaneously could expand the application of the CRISPR technology for genetic gain improvement in crops. Complex genome engineering would require efficient guided CO at each specific target site. This can be achieved by employing developmental or cell cycle-specific promoters. High efficiency multiplexed gene editing using an egg-specific promoter has been demonstrated previously in Arabidopsis[26]. Targeted DSBs introduced in cells after chromosome duplication in S or G1 phase could be repaired by recombination between sister chromatids. This type of chromosomal rearrangement is driven by the NHEJ or HDR pathways in somatic cells and would be indistinguishable from the DNA repairs resolved by the direct re-ligation at the target site. Expression of the guided nuclease at the G2 phase could potentially increase targeted CO frequency by eliminating possible

rearrangement between sister chromatids. Expressing the editing machinery at the zygote stage or splitting editing components between female and male gametes and induction of editing after pollination would be other approaches to test for increasing the efficiency to recover the desired targeted CO.

## Methods

**Plant material & plant transformation**. F1 seeds were produced by cross-pollinating two maize (*Zea mays*) inbred lines, a proprietary elite Stiff Stalk female line (Parent A) and LH244 (Parent B) by hand pollination in a field. Pollen was collected from Parent B tassels and crossed onto silks of Parent A. F1 embryo explants were transformed with *Agrobacterium tumefaciens*[27] using *EPSPS-CP4* as the plant selectable marker[28].

**Plasmids & agrobacterium strains**. *Agrobacterium tumefaciens* AB32 strain[29] carrying pCpf1_gRNA1, pCpf1_gRNA2, or pCpf1_gRNA_control plasmids with a plant selectable marker (*cp4*) were used for transformation. All three plasmids contained *LbCas12a* and gRNA expression cassettes. Editing plasmids expressed gRNA1 (pCpf1_gRNA1) or gRNA2 (pCpf1_gRNA2), a control plasmid (pCpf1_gRNA_control) expressed a "dummy" gRNA lacking the target sites in the maize genomes. Plasmid structural components driving expression of the genome-editing machinery and the selection marker are listed in Supplementary Data 5.

**Generation of BC1-F1 seeds for genotyping**. The efficiency of the nuclease-guided crossover could not be determined because of the absence of prior demonstrations. Therefore, the sample size of the regenerated plants was not calculated. In the first genome editing experiment, we produced 51 and 26 F1-T0 plants transformed with pCpf1_gRNA1 and pCpf1_gRNA_control plasmids, respectively. All regenerated F1-T0 plants were assayed for gRNA editing activity (Supplementary Fig. 1a, Supplementary Data 1). We selected and transferred to soil 42 edited and 15 control F1-T0 plants for reciprocal backcrossing due to space limitation in the greenhouse at the time of the experiment. In the second genome editing experiment, all 43 F1-T0 plants transformed with pCpf1_gRNA2 were

assayed for gRNA activity (Supplementary Fig. 1b, Supplementary Data 1) and selected for reciprocal backcrossing.

The transgenic F1-T0 generation and the parental inbred line used for reciprocal backcrossing were grown in a greenhouse with a day temperature of 29 °C and a night temperature of 21 °C, with supplemental lighting added to provide 16-h daylength. The inbred line was planted prior to, at the same time as, and after the F1 generation, to ensure overlap of pollen shed of the male population and silking of the female population. The pollination procedure was performed by hand-pollination technique. The F1-T0 plants were backcrossed with Parent B (LH244) or Parent A in the first and the second genome editing experiment, respectively. The change of the inbred parent in the backcrossing scheme was due to the ear/pollen synchronized inbred plant population available in the greenhouse at the time of the experiment.

**BC1-F1 seed chipping and genotyping**. Seed sample size was determined arbitrarily to be large enough (in thousands of samples) for statistical comparison between treatment and control. Subsets of 80 to 120 BC1-F1 seeds produced from each backcrossing event were sampled for genotyping analysis. Nondestructive sampling of BC1-F1 seed populations was performed using an automated high-throughput seed chipper[30]. The small amount of endosperm tissue was collected from each seed into 96 well plates; DNA was isolated and genotyped using quantitative endpoint TaqMan PCR assay.

Qualitative endpoint TaqMan assays were performed using TaqPath ProAmp master mix obtained from ThermoFisher Scientific (Waltham, MA USA) according to the manufacturer's protocol. Thermal cycling was performed on an Applied Biosystems (Waltham, MA USA) GeneAmp PCR system 9700 and fluorescence measurement by Tecan Spark microplate reader. TaqMan FAM- and VIC-labeled probes were obtained from ThermoFisher Scientific (Waltham, MA USA) Scientific and primers were obtained from either ThermoFisher Scientific (Waltham, MA USA) or Integrated DNA Technologies (Coralville, IA USA). The sequences of TaqMan assay primers are provided in Supplementary Data 2.

**DNA isolation**. DNA from leaf or seed endosperm tissue was extracted using a DNA-binding filter method. Leaf tissue was collected from regenerated or germinated seedlings at the V1-growth stage. Plant tissue samples were collected into 96-deep well plates, frozen, and lyophilized prior to extraction. Samples were ground by paint shaker with 3/16 in. stainless steel balls in 440 μl extraction buffer (0.1 M Tris-HCl pH 8.0, 0.05 M EDTA, 0.1 M NaCl, 1%SDS), preheated to 65 °C. Following grinding, samples were incubated at 65 °C for 45 min, followed by the addition of 135 μl of 5 M potassium acetate. After brief centrifugation, 40 μl of cleared lysate was added along with 40 μl of isopropanol to a 384-well binding filter (PALL), and the plates were centrifuged to remove the liquid waste. The bound DNA was washed with 50 μl of 70% ethanol and centrifuged. The bound DNA was eluted with 60 μl of DNA elution buffer (10 mM Tris-HCl pH 8.0, 1 mM EDTA).

**Evaluation of gRNA editing activity by Illumina sequencing**. Samples were prepared following the Illumina DNA Prep protocol (formerly FLEX) provided by Illumina (San Diego, CA USA) using in-house primers, adapted from the original Illumina primer design with unique oligonucleotides that incorporate the Illumina adapter overhang, unique indexes, and genome target-specific primers. Sequences of in-house primers are provided in Supplementary Data 2. All samples (~5 ng of total DNA) were amplified in separate reactions using Phusion High-Fidelity DNA Polymerase (ThermoFisher Scientific Waltham, MA USA). Amplicons of 424 nt or 283 nt representing gRNA target region#1 and target #2 respectively were purified using Agencourt AMPure XP beads and correctly sized products were verified using an Agilent 2100 Bioanalyzer DNA 1000 chip. Individually barcoded samples were pooled equimolarly and sequenced on an Illumina NextSeq using the NextSeq Reagent Kit 2 × 150 bp paired-end sequencing kit.

The reads from Illumina libraries were mapped to the genome reference sequences representing gRNA target regions to identify edited or wild-type reads. To analyze NGS data and evaluate gRNA editing activity, we employed proprietary software ShowEdits. The editing activity was measured as a percentage of edited reads in the total number of reads mapped to the reference sequence. Only reads with deletions but not substitution or insertions were counted as edited reads.

**Statistics and reproducibility**. We used the Chi-square test of independence to confirm statistical significance of the observed difference in recombination frequency between the control and treatment in the first genome editing experiment with gRNA1. However, the precision of the Chi-square test decreases when any of the contingency table values are low (Supplementary Table 3).

To validate the significances of the Chi-square test result, we implemented a permutation test that is not impacted by the low values in the contingency table. The following R functions were used to calculate the estimated p-value:

```
# Create the contingency table
m = matrix(c(30,1,4170,2264), nr = 2, by = T)
# Perform Pearson's Chi-squared test
chisq.test(m)
data: m
X-squared = 13.847, df = NA, p-value = 0.00024
```

```
# Perform Pearson's Chi-squared test with simulated p-value (based on 1e + 05 replicates)
chisq.test(m, sim = T, B = 1e5)
data: m
X-squared = 13.847, df = NA, p-value = 0.00028
```

To demonstrate the reproducibility of the targeted crossover in hybrid corn, we performed two independent genome editing experiments inducing chromosomal rearrangement at two different physical positions at the corn chromosome 3.

**Reporting summary**. Further information on research design is available in the Nature Research Reporting Summary linked to this article.

## Data availability
The authors declare that the data supporting the findings are available within the paper and its Supplementary Information; or are available from the corresponding author upon reasonable request. Source data underlying all figures are available in Supplementary Data 1. Biological materials including plasmids, plant tissue, or plant seeds cannot be distributed.

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

## Acknowledgements

We thank Larry Gilbertson, Lex Flagel, Sofia Brandariz Zerboni, Meenu Padmanabhan, Shilpa Swarup, Lisa Kanizay for critical reading of the manuscript. Special thanks to Giane Yanai for help with sample collection and data analysis; Samantha Schafer and Wayne Brown for assistance with seed chipping; Isaac Banks and Nancy Brumley for helping with cultivation and pollination of plants in the greenhouse; Steven Beach for assistance with Illumina sequencing, and Yuanji Zhang for help with NGS data analysis.

## Author contributions

A.K. designed and conducted the experiments, wrote the manuscript, A.S. and V.S performed plant transformation, S.H. performed TaqMan assay design and helped with genotyping analysis, B.L. produced the hybrid plants, T.B. performed gRNA activity validation, C.A., M.G., R.L, and S.Y. contributed to experimental design, manuscript editing, and scientific discussions.

## Competing interests

All authors declare potential competing interests. All authors are employed by Bayer Crop Sciences.
