## [Peer Review File · Communications Biology]

Reviewers' comments:

Reviewer #1 (Remarks to the Author):

The authors used a hybrid between two inbred maize lines and subjected it to CRISPR editing to determine if exchange could be produced between homologous chromosomes. Backcrosses to the two different parents were analyzed via sequencing to determine whether recombination had been induced. The claim is that there is an increase of about 18 fold with the editing construct present compared to the control recombination frequency in the same interval. Sequencing of the presumed editing mediated recombinants indicates changes in nucleotides, which would be consistent with editing. The presumptive recombination events were made homozygous.

What this reviewer does not understand is the following: Embryos of hybrid plants were transformed with the editing machinery and then grown to plants for backcrosses. If the editing occurs early in this process then the plants regenerated would have large sectors carrying the recombination events. Thus, they would be present in high frequency in the backcross progeny. There is not a lot of clarity in the manuscript about the sampling procedure. The manuscript does note some mosaicism from the same event; for this to occur the editing must occur during the development of the regenerating plant, which is certainly possible, but a higher frequency would still be expected from somatic sectors leading to the ear and tassel. In a normal embryo, there are only one or two cells that lead to the ear and perhaps about a dozen that lead to the tassel. The number of cells in a somatic embryo leading to the ear and tassel is not clear. The citation to Dumas and Mogensen does not include any information that illuminates this point.

In order to obtain a somatic recombination event, a CRISPR cut must occur in the two homologues simultaneously and a repair event occurs that switches the products between the homologues if NHEJ mediates the repair. It is likely that a complicating factor is that there will be edits occurring on one or the other homologue independently, some of which will stop the editing process because of the change in nucleotide sequence. It is stated that both gRNAs show strong editing activity; it would be good to compare in the text this activity to the frequency of guided CO. Perhaps the low frequency is due to the need for independent events to occur on the two homologues and for an exchange to occur. If this fits with the authors understanding, they need to lead the reader through the logic.

Reviewer #2 (Remarks to the Author):

The manuscript described studies demonstrating Cas12a-mediated targeted (somatic) chromosomal crossover (CO) in transgenic hybrid maize plants. The results showed heritable crossover in F1 plants when Cas12a transgene was segregated away. The frequency of LbCas12a-mediated CO is significantly higher than control natural CO in the same intervals in the progeny of 3 different events. The results are significant since plant breeding efficiency is often limited by chromosomal recombination rate, especially in low recombination regions. Previously, other researchers have shown that Cas9 can mediate enhancement of targeted crossover in tomato. This is another demonstration of targeted crossover enhancement using CRISPR technology in an important food crop.

Overall, the results are solid and support conclusions. Also, the manuscript is well written. Here are a few points for consideration in the revision.

- 1) Page 2, line 13, Figure 2A, this is the first time any figure is shown. Can it be made into part of the Figure 1?
- 2) Page 4, line 5, Figure 2C and 2D shows 13/31 BC1-F1 CO seeds are Cas12a negative, not 14/30?
- 3) Page 4, line 5, add "with CO" between "from edited plants" and "were LbCas12a negative".
- 4) Page 4, line 12, add Fig. 2C and 2D after Fig.2B since Fig 2B does not talk about Event 4?
- 5) Page 4, line 19, "Even 1" is a typo of "Event 1"
- 6) Page 6, line 8, Figure 3B should be Figure 3C.
- 7) Page 6, line 15, change several to "3" to be more specific.

- 8) Page 7, line 5, the data cannot exclude the possibility that CO happened during meiosis, especially for Event 2 and 3, right? I agree that CO in Event ZM_S22440456 were most likely from somatic recombination.
- 9) Page 7, line 14 and 17 "cp4" is not quite correct as selectable marker. Is it better to change it to "EPSPS-CP4" or "aroA-CP4"?
- 10) Page 7, line 16, the second "pCpf1-gRNA2" should be "pCpf1_gRNA_control".
- 11) Page 8, line 16, 17, 18 and 19, add manufactures' locations.
- 12) It looks like the numbering of reference is not following the order of their appearance starting from page 4 (line 15, refence 18 and 19). Can they be rearranged?
- 13) Figure 2B, the number of events (42 and 15) are different from the ones in Suppl Table 1 where Event number of pCpf1_gRNA1 are 47, not 42, pCpf1-gRNA_control are 18 not 15. Please explain or correct.
- 14) Suppl Fig 1. Red stars..... "were"... should be "where".
- 15) Any explanation for the low occurrence of events with somatic crossover? Further discussions on how to improve somatic crossover would be interesting, e.g. is it possible that somatic crossover requires the cells to be in a particular phase of the cell cycle?

Reviewer 1 Comment	Response
What this reviewer does not understand is the following: Embryos of hybrid plants were transformed with the editing machinery and then grown to plants for backcrosses. If the editing occurs early in this process then the plants regenerated would have large sectors carrying the recombination events. Thus, they would be present in high frequency in the backcross progeny. There is not a lot of clarity in the manuscript about the sampling procedure. The manuscript does note some mosaicism from the same event; for this to occur the editing must occur during the development of the regenerating plant, which is certainly possible, but a higher frequency would still be expected from somatic sectors leading to the ear and tassel. In a normal embryo, there are only one or two cells that lead to the ear and perhaps about a dozen that lead to the tassel. The number of cells in a somatic embryo leading to the ear and tassel is not clear. The citation to Dumas and Mogensen does not include any information that illuminates this point.	1. We agree with the reviewer that high frequency recombination should be expected in the backcross progeny if active gRNAs induce targeted recombination early in the development. We demonstrated this result in our second experiment with gRNA2. To address this comment, we included the formal Discussion section in the manuscript. In the discussion, we compared the results of two genome editing experiments and provided explanations of the mosaicism and lower frequency of the targeted CO observed in the genome editing experiment with gRNA1. Please see page 18, lines 4 to 25 and page 19 lanes 1-20, and page 20 lanes 5-20. We also provided additional experimental evidence of targeted CO in early germinal precursor cell in the second experiment with gRNA2 showing absence of mosaic editing patterns in BC1F1 progeny. See Figure 4 D-F. We also confirmed the stable transmission of targeted CO induced in the presence of gRNA2 to the BC1F2 progeny and provided genotyping analysis as supplementary table: Genotyping_BC1F2_gRNA2.xlsx 2. In our study, we sampled F1-T0 plants right after plugging collecting leaf disks to assess gRNA editing activity. At this stage we could only evaluate gRNA editing or cutting efficiency but not the chromosome recombination efficiency. BC1-F1 seeds represent the earliest stage where targeted CO could be identified. Added to Material & methods: Leaf tissue from the regenerated F1-T0 plants were collected right after plugging at V1-growth stage. See page 19: line 11-25 & page 20 page 1-4
In order to obtain a somatic recombination event, a CRISPR cut must occur in the two homologues simultaneously and a repair event occurs that switches the products between the homologues if NHEJ mediates the repair. It is likely that a complicating factor is that there will be edits occurring on one or the other homologue independently, some of which will stop the editing process because of the change in nucleotide sequence. It is stated that both gRNAs show strong editing activity; it would be good to compare in the text this activity to the frequency of guided CO. Perhaps the low frequency is due to the need for independent events to occur on the two homologues and for an exchange to occur. If this fits with the authors understanding, they need to lead the reader through the logic.	In our study we did not observe strong correlation between gRNA editing activity and frequency of the targeted CO. We discussed the details in the Discussion section. Please see page 9 lanes 1- 19

	Reviewer 2 Comment	Response
1	Page 2, line 13, Figure 2A, this is the first time any figure is shown. Can it be made into part of the Figure 1?	Figure 2A is made to Figure 1. The figure order is adjusted accordingly
2	Page 4, line 5, Figure 2C and 2D shows 13/31 BC1-F1 CO seeds are Cas12a negative, not 14/30?	Corrected
3	Page 4, line 5, add “with CO” between “from edited plants” and “were LbCas12a negative”.	Corrected
4	Page 4, line 12, add Fig. 2C and 2D after Fig.2B since Fig 2B does not talk about Event 4?	Corrected
5	Page 4, line 19, “Even 1” is a typo of “Event 1”	Corrected
6	Page 6, line 8, Figure 3B should be Figure 3C.	Corrected
7	Page 6, line 15, change several to “3” to be more specific.	Corrected
8	Page 7, line 5, the data cannot exclude the possibility that CO happened during meiosis, especially for Event 2 and 3, right? I agree that CO in Event ZM_S22440456 were most likely from somatic recombination.	We agree with the reviewer, we can't exclude that the targeted COs observed in the first genome experiment using gRNA1 were introduced during meiosis. Added in text: “We assume that the targeted CO in the experiment with gRNA1 could be induced after subepidermal cells of the ovule primordia developed into archesporial cell that represent precursors for MMC (Megaspore Mother Cell), or maybe even later when MMC undergoes meiosis (23)”. Please see page 19 lanes 5-10
9	Page 7, line 14 and 17 “cp4” is not quite correct as selectable marker. Is it better to change it to “EPSPS-CP4” or “aroA-CP4”?	Corrected, changed to EPSTS-CP4
10	Page 7, line 16, the second “pCpf1-gRNA2” should be “pCpf1_gRNA_control”.	Corrected
11	Page 8, line 16, 17, 18 and 19, add manufactures' locations.	Addresses added as (City/State/Country)
12	It looks like the numbering of reference is not following the order of their appearance starting from page 4 (line 15, refence 18 and 19). Can they be rearranged?	Corrected. References rearranged according to reviewer recommendations

13	Figure 2B, the number of events (42 and 15) are different from the ones in Suppl Table 1 where Event number of pCpf1_gRNA1 are 47, not 42, pCpf1-gRNA_control are 18 not 15. Please explain or correct.	The paragraph was added to Material & Methods: We produced 51 and 27 F1-T0 plants transformed with pCpf1_gRNA1 and pCpf1_gRNA_control plasmids, respectively. All regenerated F1-T0 plants were assayed for gRNA editing activity (Supplementary Figure 1A). We selected and transferred to soil 42 edited and 15 control F1-T0 plants for reciprocal backcrossing due to space limitation in the greenhouse at the time of experiment. In the second genome editing experiment, all produced 43 F1-T0 plants transformed with pCpf1_gRNA2 were assayed for gRNA activity (Supplementary Figure 1B) and selected for reciprocal backcrossing. Please see page 22 lanes 1 - 6
14	14) Suppl Fig 1. Red stars..... “were” ... should be “where”.	Corrected
15	Any explanation for the low occurrence of events with somatic crossover? Further discussions on how to improve somatic crossover would be interesting, e.g. is it possible that somatic crossover requires the cells to be in a particular phase of the cell cycle?	Added a paragraph to the discussion section to address the reviewer comment: This complex engineering design would require efficient guided CO at each specific target site. This can be achieved by employing the developmental or cell cycle specific promoters. High efficiency multiplexed gene editing using egg specific promoter has been demonstrated previously in Arabidopsis (29). Targeted DSBs introduced in cells after chromosome duplication in S or G1 phase could be repaired by recombination between sister chromatids. This type of the chromosomal rearrangement is driven by NHEJ or HDR pathways in somatic cells and would be indistinguishable from the DNA repairs resolved by the direct relegation at the target site. Expression of the guided nuclease at the G2 phase could potentially increase targeted CO frequency by eliminating possible rearrangement between sister chromatids. Expression editing machinery at zygote stage or splitting editing components between female and male gametes and induction of editing after pollination would be other designs to test for efficiency improvement of the targeted CO. Please see page 20 lanes 21-25 & page 21 page 1 -8

REVIEWERS' COMMENTS:

Reviewer #1 (Remarks to the Author):

The existence of somatic COs was previously well established. In the revision, the discussion of the developmental aspects to explain the frequency of recovery now makes more sense. There is a need for some smoothing of the language, specifically the addition of the article "the" in many places--too many to enumerate.

Reviewer #2 (Remarks to the Author):

The revised manuscript has addressed all my questions. It now flows much better. Also, the updated discussion greatly helps with the manuscript too. Here are a couple more minor edits to help with the manuscript,

- 1) Page 11, Figure 4B, the label for X axis, "Cpf1-negative BC1-F1 events" should be "Cpf1-negative BC1-F1 plants". These plants are progeny "plants" of the "events". The same for Fig. 4C and 4F, in tables' 1st row, "BC1-F1 events #" should be "BC1-F1 plants #".
- 2) Page 12, 14, Fig 4, Table C & F, please replace "Editing" with the construct number, e.g. pCpf1-gRNA1,2, 3 & pCpf1-gRNA2. Also, in Fig 4 legends, replace "lbCpf1" with "LbCpf1" in 2 places.
- 3) Suppl Figure 1, please label event # for the individual asterisk. It is important for the reader to know the editing status of the individual T0 event in whether it is a monoallelic or biallelic editing. It is hard to figure out if pCpf1-gRNA Event 2 or Event 3 is a biallelic edited event. Alternatively, add in the last line of page 4 that Event 1, 2 and 3 have mono-, bi- and mono-allelic editing.

Reviewer 1 Comment	Response
The existence of somatic COs was previously well established. In the revision, the discussion of the developmental aspects to explain the frequency of recovery now makes more sense. There is a need for some smoothing of the language, specifically the addition of the article "the" in many places--too many to enumerate.	The manuscript was edited to address grammar issues pointed out by the reviewer.

	Reviewer 2 Comment	Response
1	The revised manuscript has addressed all my questions. It now flows much better. Also, the updated discussion greatly helps with the manuscript too. Here are a couple more minor edits to help with the manuscript, 1) Page 11, Figure 4B, the label for X axis, "Cpf1-negative BC1-F1 events" should be "Cpf1-negative BC1-F1 plants". These plants are progeny "plants" of the "events". The same for Fig. 4C and 4F, in tables' 1st row, "BC1-F1 events #" should be "BC1-F1 plants #".	X axis labels on Figure 4B and 4 D were changed to LbCas12a-negative BC1-F1 plans. Column names were changed to "BC1-F1 plants #" in Tables 5 and 6
2	Page 12, 14, Fig 4, Table C &F, please replace "Editing" with the construct number, e.g. pCpf1-gRNA1,2, 3 &pCpf1-gRNA2. Also, in Fig 4 legends, replace "lbCpf1" with "LbCpf1" in 2 places.	Renamed column "Editing" to "Construct" and provided constructs names: pCpf1-gRNA1 & pCpf1-gRNA2 in Table 5 and 6. Corrected legend labels to LbCas12a in Figure4b and Figure4d
3	Suppl Figure 1, please label event # for the individual asterisk. It is important for the reader to know the editing status of the individual T0 event in whether it is a monoallelic or biallelic editing. It is hard to figure out if pCpf1-gRNA Event 2 or Event 3 is a biallelic edited event. Alternatively, add in the last line of page 4 that Event 1, 2 and 3 have mono-, bi- and mono-allelic editing.	Labeled individual asterisk on Suppl Figure1a with corresponding event # as presented in the manuscript. Provided a note in the figure legend: Number 1, 2, and 3 represent described in the article the recombinant events #1, #2, and #3, respectively. It is impossible to conclude that any F1-T0 events are mono or biallelic because both parental targeted regions are completely homologous. To address reviewer remark, we included the following paragraph to the Result section: "Both gRNAs

showed strong editing activity in the F1-T0 plants. An average number of reads with the DSB editing patterns was 87% and 80% at the gRNA1 and gRNA2 target sites, respectively (Supplementary Figure 1a and Figure 1b). All assayed F1-T0 plants contained multiple patterns of DSB editing showing chimerism frequently observed in primary (T0) plant transformants (11). Distribution of the editing patterns in the F1-T0 plants are shown in four examples (Supplementary Figure 2a and Figure 2b). We cannot definitively conclude that the editing observed in the F1-T0 plants is mono or biallelic because of the absence of the allelic polymorphism at the gRNA1 and gRNA2 target sites. However, we can assume that both parental alleles were edited in plants where editing efficiency was above 50%.”. We also provided Supplementary Figure 2 a and 2b to show distribution of the editing patterns in Event1, Event2, and Event3.